# The First *Schaalia* (Formerly *Actinomyces*) *Canis*-Related Osteomyelitis Requiring Surgical Intervention

**DOI:** 10.3390/idr17040094

**Published:** 2025-08-04

**Authors:** Patrick Nugraha, Tzong-Yang Pan, Paul Di Giovine, Nigel Mann, William Murphy

**Affiliations:** Department of Surgery-Plastic and Reconstructive Surgery, The Northern Hospital, Northern Health, Epping, VIC 3076, Australia; patricknugraha1@gmail.com (P.N.); tzong.yang.pan@gmail.com (T.-Y.P.); pdigi0012@gmail.com (P.D.G.); nigel.mann@nh.org.au (N.M.)

**Keywords:** *Schaalia canis*, *Actinomyces canis*, osteomyelitis, zoonotic infection, dog bite

## Abstract

*Schaalia canis* is a Gram-positive, facultatively anaerobic, rod-shaped bacterium originally isolated from the mucosa and skin of dogs. While it is a part of the normal canine oral flora, it has rarely been implicated in human disease, with only one prior case of cellulitis reported following a dog bite. **Case Presentation:** We present the case of a 57-year-old immunocompetent man who developed osteomyelitis of the left index finger following a delayed presentation after a dog bite. Despite initial conservative management with empirical oral antibiotics, the infection progressed, eventually requiring surgical debridement and the terminalisation of the finger at the proximal interphalangeal joint. Cultures from intraoperative bone specimens yielded the growth of *Schaalia canis*, with no other pathogenic organisms identified on the extended culture. **Conclusions:** This is the first documented case of *Schaalia canis*-associated osteomyelitis in a human and the first to necessitate a surgical intervention, expanding the known clinical spectrum of this organism. This case underscores the risks of delayed intervention in polymicrobial animal bite wounds and highlights the emerging role of *Schaalia* species as opportunistic zoonotic pathogens, particularly in the setting of deep, refractory infections.

## 1. Introduction

The *Actinomycetaceae* family comprises a diverse group of Gram-positive, filamentous, non-spore-forming, rod-shaped bacteria that are widely distributed in nature. These organisms are commonly found in soil and water and form part of the normal commensal flora in humans and animals, particularly within the oral cavity, gastrointestinal tract, and genitourinary mucosa [1,2]. Among the genera within this family, *Actinomyces* has been one of the more extensively studied, known for its capacity to transition from a benign commensal to an opportunistic pathogen. Invasive *Actinomyces* infections are characteristically chronic and indurated, often forming draining sinus tracts and dense fibrotic tissue, particularly when mucocutaneous barriers are disrupted [1]. These infections can mimic neoplastic or granulomatous diseases, posing significant diagnostic challenges [1,2].

Despite the clinical familiarity with *Actinomyces* species, the full diversity within this family has only begun to be appreciated recently. Advancements in molecular diagnostics, such as 16S rRNA gene sequencing and whole-genome analysis, have enabled more precise taxonomic categorization and have uncovered the phylogenetic heterogeneity previously masked by phenotypic similarity [3]. As a result, several species formerly grouped under *Actinomyces* have been reassigned into novel genera. One such taxonomic revision led to the establishment of the genus *Schaalia* in 2018, which encompasses species formerly classified under *Actinomyces*. *Actinomyces canis*, originally isolated from the mucous membranes and skin of dogs in 2000, was subsequently reclassified as *Schaalia canis*, reflecting its distinct phylogenetic lineage [3,4].

Despite its presence within the canine microbiota, clinical reports involving *S. canis* in humans remain exceedingly rare. To date, only a single documented human infection caused by *S. canis* exists in the literature. That case involved lower-limb cellulitis in a paediatric patient following a dog bite, which was resolved with intravenous antibiotic therapy alone and did not require surgical intervention [5]. In contrast, this report describes the first known human case of *S. canis*-associated osteomyelitis, following a delayed presentation after a dog bite injury. The infection progressed despite multiple courses of empirical oral antibiotics, ultimately requiring surgical debridement and partial digit amputation.

This case report aims to contribute to the limited but growing body of literature regarding *S. canis* as a zoonotic pathogen. By detailing this rare presentation, we aim to raise clinical awareness of this underrecognized organism and emphasize the importance of the early diagnostic consideration of uncommon pathogens in persistent refractory infections.

## 2. Case Presentation

A 57-year-old immunocompetent man with no use of immunosuppressive medications or biologic agents, and no chronic comorbidities known to impair immune function, had sustained a dog bite injury to his left index finger. The bite resulted in a puncture wound over the volar aspect of the distal phalanx. He initially presented to a regional general practice (GP) clinic, where the wound was irrigated with normal saline, dressed, and managed conservatively with a five-day course of oral amoxicillin–clavulanic acid (875 mg every 12 h), which was the recommended empirical therapy for localised bite injuries based on local guidelines [6]. After finishing this course, the patient still noted the progressive worsening of erythema, swelling, and pain, accompanied by the reduced range of motion of the finger.

Over the next two weeks, he re-presented himself to the GP clinic multiple times due to persistent and worsening symptoms. In response to this clinical deterioration, he was started on two additional oral empirical antibiotic courses: trimethoprim/sulfamethoxazole (160 mg/800 mg every 12 h) and metronidazole (400 mg every 8 h) for five days, followed by cefalexin (500 mg every 6 h) for another five days. Despite these additional courses of antibiotics, the infection persisted, prompting further investigation and an escalation of care.

Intravenous antibiotics were recommended at this stage due to the failure of repeated oral treatments. However, the patient declined hospital admission because of personal and logistical challenges. Despite counselling on the risks of an untreated infection, including progression and the possible loss of the affected digit, efforts to persuade him were unsuccessful. The patient did, however, agree to outpatient imaging to assess the extent of the infection. A plain radiograph performed approximately 1 month post-injury demonstrated soft tissue swelling and clear bony erosions involving both the distal and middle phalanges, raising a high suspicion for osteomyelitis (Figure 1). An urgent referral was made to the plastic and reconstructive surgery unit; however, due to ongoing personal circumstances, the patient further delayed his presentation.

Six weeks after his initial dog bite injury, he was eventually presented to the emergency department at a tertiary hospital. Tetanus prophylaxis was administered on admission. On assessment, the patient was systemically well and afebrile. Laboratory investigations showed mildly elevated C-reactive protein (CRP) of 7 mg/L (normal reference range: 0–5 mg/L), while the white cell count was 10.8 × 10^9^/L (normal reference range: 4.5–11.0 × 10^9^/L). The examination revealed severe dactylitis of the left index finger, characterized by erythema, oedema, and a dusky appearance of the middle and distal phalanx (Figure 2A). The digit was markedly tender, with a reduced active and passive range of motion. Additionally, two puncture wounds with purulent discharge were identified over the distal volar aspect of the left index finger (Figure 2B), with surrounding induration and inflammation extending proximally.

A percutaneous bone biopsy along with targeted IV antibiotic therapy were considered for the conservative management of the osteomyelitis; however, the degree of the bony destruction was very significant, with a near-complete mid-shaft transverse destruction. This would have required long-term antibiotics, future bone grafting, and a significant amount of hand therapy rehabilitation, which the patient opted against. The surgical team determined that debridement and terminalization of the affected digit were necessary to achieve effective source control and minimize the risk of local spread and systemic complications (Figure 3). The procedure was performed within 24 h of the initial admission.

Intraoperative findings revealed the extensive necrosis of the bone and surrounding soft tissue, consistent with osteomyelitis. Bone tissue samples were collected for Gram-staining and a bacterial culture. A microscopic examination of the Gram-stained preparation revealed Gram-positive coccobacilli (Figure 4).

To facilitate the isolation of both fastidious and non-fastidious organisms, the intraoperative tissue samples were placed on three separate media culture plates: 1× horse blood agar (HBA) and 1× chocolate horse blood agar (CHBA) under aerobic conditions with 5% CO_2_ and 1× HBA under anaerobic conditions. All cultures were maintained at 37 °C. After seven days of incubation, the bacterial growth was most prominent on the aerobic plates (Figure 5). The identification was confirmed as *Schaalia canis* using matrix-assisted laser desorption/ionization time-of-flight mass spectrometry (MALDI-TOF MS; Bruker ^®^ MBT, Dorevitch Pathology, Melbourne, Australia). No additional pathogenic organisms were isolated. Antibiotic susceptibility testing was not performed in this case, in line with standard practices, as *Schaalia* and *Actinomyces* species are generally susceptible to beta-lactam antibiotics [7]. However, had there been a lack of clinical improvement with the beta-lactam therapy, further antibiotic sensitivity testing would have needed to be pursued to guide targeted treatment.

Postoperatively, following a consultation with the infectious diseases team, intravenous meropenem (1 g every 8 h) was initiated due to the high polymicrobial risk associated with animal bite injuries and the severity of the established infection. The patient responded well, with a progressive reduction in swelling, improved pain control, and no systemic signs of infection throughout his hospital stay. After one week of inpatient therapy, he was discharged to complete a six-week course of oral amoxicillin–clavulanic acid (875 mg every 12 h). The subsequent outpatient follow-up at 1 and 6 weeks demonstrated the complete resolution of local symptoms with no evidence of recurrence or ongoing inflammation. No further surgical intervention was required after the completion of antibiotics.

## 3. Discussion

This case represents the second documented human infection caused by *Schaalia canis* associated with dog bite injuries. The previously reported case involved lower-limb cellulitis in a 12-year-old, which was managed early with a single course of intravenous amoxicillin–clavulanic acid and did not require any surgical intervention [5]. In contrast, our patient experienced a delayed presentation of up to six weeks, allowing the infection to progress to osteomyelitis, ultimately necessitating surgical debridement and the amputation of the digit. This case underscores the critical role of early interventions in post-traumatic infections and highlights the potential for zoonotic pathogens such as *S. canis* to cause refractory infections even in otherwise healthy, immunocompetent individuals.

Dog bite injuries are polymicrobial by nature, typically involving a complex combination of aerobic and anaerobic bacteria [8]. Commonly implicated pathogens include *Pasteurella multocida, Pasteurella canis, Streptococcus* spp., *Staphylococcus* spp., as well as anaerobes such as *Fusobacterium* spp. and *Prevotella* spp. [8,9]. Empirical antibiotic therapy with amoxicillin–clavulanate is widely recommended because of its broad-spectrum activity against many of these organisms [9,10]. However, as illustrated in this case, the initial administration of empirical oral antibiotics proved inadequate in resolving the infection- likely due to both the delayed presentation, which permitted a higher bacterial burden to accumulate, and the inherently polymicrobial nature of dog bite wounds [11]. While oral amoxicillin–clavulanate may offer an appropriate empirical coverage for superficial animal bites, it may be insufficient for deeper or more established infections [10,12].

Although not specific to *S. canis*, a 2006 study by Thulin et al. investigating *Streptococcus pyogenes* infections provided relevant examples of the challenges associated with treating deep tissue infections characterized by high bacterial loads. In this study, snap-frozen biopsies taken from patients with necrotizing fasciitis and severe cellulitis revealed that viable *S. pyogenes* could be found not only in the centre of the infection but also in the surrounding tissue that appeared clinically unaffected. In situ imaging showed a strong correlation between the bacterial density and the severity of the soft tissue inflammation, with over 70% of severely affected areas containing high bacterial loads, whereas distal regions had lower loads [13].

Notably, viable bacteria were still present in tissue samples collected up to 20 days after the diagnosis and after the initiation of intravenous antibiotic therapy, including beta-lactams and clindamycin [13]. This persistence suggests that even appropriately selected intravenous antibiotics may fail to fully eradicate pathogens in high-burden or poorly perfused tissues. These findings highlight the limitations of antibiotic therapy alone in managing deep or delayed infections. Clinicians should remain vigilant in such cases, maintaining a low threshold for early imaging, the escalation to intravenous therapy, or surgical referrals [13,14,15].

An additional consideration is the importance of timely and thorough wound management. While no direct comparison was available in this case, prompt wound irrigation and decontamination at the time of injury have been shown to significantly reduce infection rates and should be standard in bite wound care progression [16,17]. Moreover, delays in receiving appropriate care, whether due to patient factors or health system barriers, can profoundly impact clinical outcomes, as evidenced in this case.

While *Schaalia* species are generally considered part of the normal mucosal microbiome, increasing evidence suggests they may act as opportunistic pathogens in both veterinary and human contexts [2,5]. Notably, the only other detailed account of a *S. canis*-related pathology involved a case of mandibular actinomycosis in a grey four-eyed opossum (*Philander opossum*) described in the veterinary literature [3]. In that report, the infected opossum presented with severe unilateral mandibular swelling and was initially treated with marbofloxacin, a broad-spectrum fluoroquinolone antibiotic commonly used to treat bacterial infections in the veterinary population [18]. Despite treatment, the opossum died within 24 h, and the post-mortem examination revealed extensive soft tissue infection and septicaemia. The culture and whole genome sequencing identified a *Schaalia* species with a 97% 16S rRNA gene sequence similarity to *S. canis*. This case in a non-human mammal parallels our human case in several respects, particularly in terms of the bony tissue involvement, progression to severe infection, and resistance to the initial empirical antibiotic therapy.

This veterinary case report further supports the hypothesis that *Schaalia* spp. may be opportunistic pathogens with potential to cause fulminant infections under the right conditions. The findings also implied a possible underestimation of their pathogenic potential due to challenges in routine microbiological identification and the limited representation in the clinical literature. Given the scarcity of documented cases and the limitations of current diagnostic modalities, further investigation into the pathogenic potential, antimicrobial susceptibility, and clinical course of *S. canis* is warranted.

Understanding its role within polymicrobial infections is particularly important, as it may be overshadowed by more commonly recognized co-pathogens. Advancements in microbial sequencing and the routine adoption of MALDI-TOF may enhance recognition and inform better clinical decision-making. The continued reporting and genomic characterization of such cases are essential in improving diagnostic accuracy, enhancing our understanding of the pathogen’s epidemiology and virulence factors, informing antimicrobial stewardship, and ultimately optimizing patient outcomes. As awareness grows and diagnostic techniques advance, more cases are likely to be identified, helping to better define the clinical relevance and management strategies for *Schaalia canis* and other emerging zoonotic pathogens.

## 4. Conclusions

*Schaalia canis* is a rare but emerging zoonotic pathogen capable of causing serious infections, even in immunocompetent individuals. This case of *S. canis*-associated osteomyelitis in a 57-year-old man following a dog bite represents the first documented instance requiring surgical intervention, thereby expanding the known clinical spectrum of this organism. Typically, a part of the canine oral microbiome, *S. canis,* can act as an opportunistic pathogen when host defences are compromised, particularly in the setting of trauma, inadequate initial care, or delayed medical attention.

This case highlights the importance of early and aggressive interventions in animal bite injuries, which are frequently polymicrobial and may harbor atypical or underrecognized organisms. Empirical therapy should provide broad-spectrum coverage, but clinicians must be prepared to escalate treatment based on the clinical response and culture results. Imaging and surgical consultation should not be delayed when an infection is suspected to extend to deep tissues or bone, as timely action is critical for optimal patient outcomes.

Furthermore, the parallels between this case and the veterinary literature—particularly a recent report of a novel *Schaalia* species causing a fatal mandibular infection in a grey four-eyed opossum—suggest that *Schaalia* species may have a broader pathogenic potential across species than previously thought. Clinicians should maintain a high index of suspicion for rare or novel pathogens in cases of persistent infection following animal exposure. Continued reporting and genomic surveillance will be essential in enhancing our understanding and guiding future therapeutic strategies for *S. canis* and other underrecognized zoonotic pathogens.

## Figures and Tables

**Figure 1 idr-17-00094-f001:**
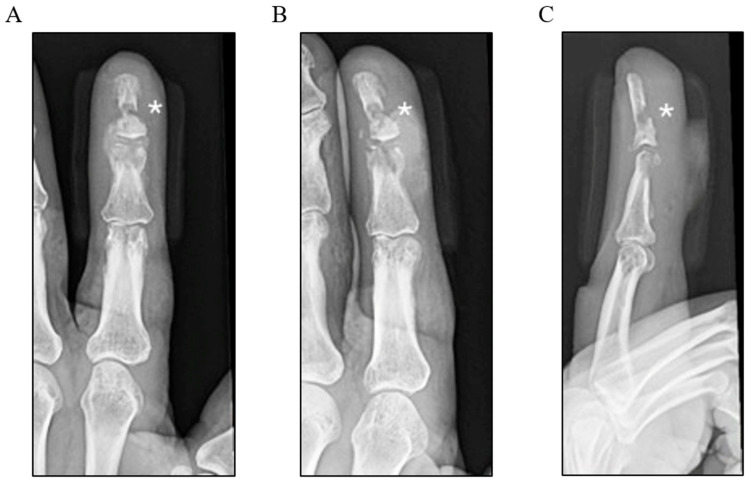
Three-view plain X-ray radiographs of the left index finger: (**A**) posteroanterior, (**B**) oblique, and (**C**) lateral views, labelled with * highlighting the bony erosions of the distal and middle phalanges.

**Figure 2 idr-17-00094-f002:**
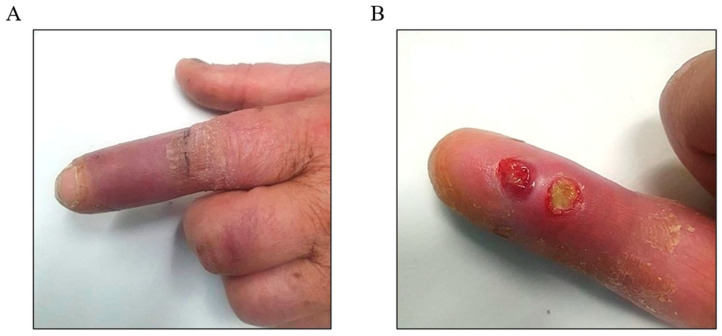
The left index finger wound 6 weeks post-injury: (**A**) The erythematous and dusky appearance up to the proximal interphalangeal joint. (**B**) The discharging wound on the volar aspect of the distal phalanx.

**Figure 3 idr-17-00094-f003:**
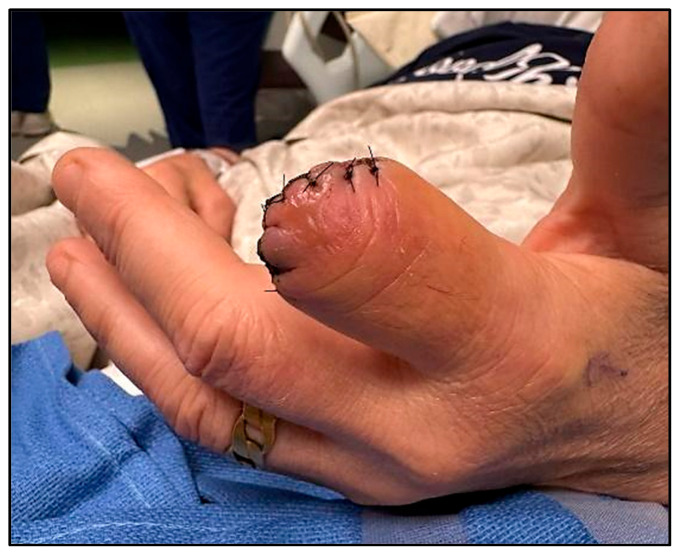
The left index finger post-surgical debridement and the terminalization up to the proximal interphalangeal joint.

**Figure 4 idr-17-00094-f004:**
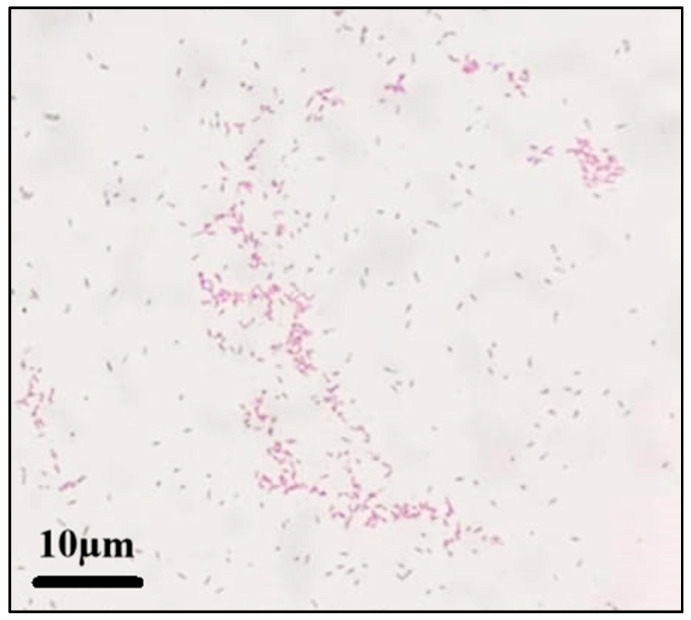
The Gram-staining of bacteria isolated from the intraoperative bone tissue sample under a 100× microscopic magnification.

**Figure 5 idr-17-00094-f005:**
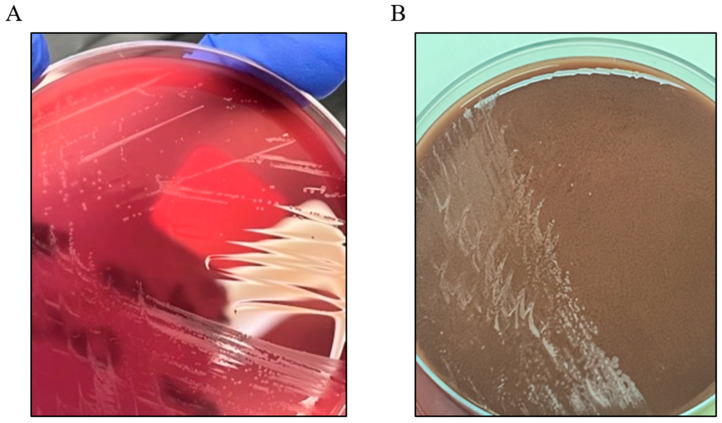
Growth of Schaalia canis colonies on (**A**) HBA and (**B**) CHBA, both incubated under aerobic conditions (5% CO_2_, 37 °C).

## Data Availability

The data presented in this study are available upon request from the corresponding author.

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
