# Peer review of "The First Schaalia (Formerly Actinomyces) Canis-Related Osteomyelitis Requiring Surgical Intervention"

_2036-7449, 2025, doi:10.3390/idr17040094_

Round 1

Reviewer 1 Report

Comments and Suggestions for Authors
  • Throughout the manuscript, replace flora with microbiota.
  • First time mention Schaalia canis, then S.canis (as in line 165).
  • The introduction repeats information, edit the section:

In contrast, the current case report describes the first known human case of Schaalia canis-associated osteomyelitis, following a delayed presentation after a dog bite injury. The infection advanced despite multiple courses of empirical oral antibiotic therapy and ultimately necessitated surgical debridement and partial digit amputation.

This case report aims to contribute to the limited but growing body of literature regarding Schaalia canis as a zoonotic pathogen. We present a rare case of Schaalia canis-related osteomyelitis requiring surgical management to raise clinical awareness of this underrecognized organism. The report underscores the potential complications of delayed treatment in dog bite injuries and highlights the importance of early diagnostic consideration for uncommon or emerging pathogens in cases of persistent, refractory, or atypical infections.

  • Lines 160-161, specify which antibiotics.
  • Line 183: in situ - italics
  • Lines 204-210: similar information is in the introduction, please remove or edit this section.
  • A summary in the discussion is not necessary when the manuscript includes a conclusion. Combine and edit these sections, some information is repeated.

Reviewer 2 Report

Comments and Suggestions for Authors

A very interesting and illustratively presented case. It would be helpful to clarify after how many days post-injury the patient sought medical attention, as it is mentioned that it was delayed. Why was amoxicillin-clavulanic acid prescribed for only 5 days? Was the deterioration observed during the course of treatment or after it was discontinued? Why was the possibility of a percutaneous bone biopsy not considered, along with targeted antibiotic therapy for osteomyelitis with good bone penetration in an attempt to preserve the finger? Was rabies and tetanus prophylaxis administered? How was the patient’s immunocompetence verified?

Reviewer 3 Report

Comments and Suggestions for Authors

The described Schaalia infection in thimmunocompetent man gives a novel insight into new exotic zoonotic bacterial diseases. Schaalia species may belong to the normal oral microbiome and may serve as a contributor to opportunistic infections. Due to the lack of current literature, more insights and improved knowledge about Schaalia spp. and their pathogenicity will be useful to choose appropriate therapy regimens and improve the treatment success rate. This is an interesting and relevant study given the dangers of zoonotic pathogens. The biggest limitation of this study is the lack of information about the antimicrobial sensitivity.

Please change the citation format from: … barriers are disrupted1 to barriers are disrupted [1]

References should be described as follows, depending on the type of work:

  • Journal, Articles:
    1. Author 1, A.B.; Author 2, C.D. Title of the article. Abbreviated Journal Name YearVolume, page range.

Reviewer 4 Report

Comments and Suggestions for Authors

Thank you for the opportunity to review the above manuscript.

Add a clear statement in the introduction section outlining the specific objectives of the study. It is important to emphasize what this research aims to evaluate or discover. Additionally, highlight how this study differs from or builds upon previous work in the field, underlining its potential impact, innovative aspects, or relevance to current scientific or clinical challenges. This will help the reader understand the significance and contribution of the study within the broader context of existing literature.

Carefully review the Materials and Methods section and add the brand, model, and country of origin for all reagents and equipment used, where applicable.

In the conclusion section, add a brief but comprehensive paragraph that highlights the main findings of the study. Emphasize how these results contribute to the understanding or management of the condition being studied, and discuss their potential implications across various medical fields. Additionally, include a reflection on the broader relevance of the findings, and outline the key lessons or takeaways that may guide clinical practice, future research, or decision-making for healthcare professionals and researchers alike. This will help underscore the value and applicability of the study for a diverse readership.

Review whether all references comply with the format required by the journal.

The background to the problem presented sufficient information from current literature to
frame the argument. The Problem was supported from a broad perspective. The authors state that, the beneficial effects of physical activity on health can reduce the risk of obesity and play a role in the prevention of obesity-related chronic diseases. However, it is not yet fully established which intensity, duration, frequency, type, and context is most important for PA. The study, therefore, set out to examine the benefits, intensity, duration, frequency, type of physical activity that are most important for the prevention of obesity.
The purpose statement or main objective for conducting the study was NOT clearly and
explicitly stated.
A quantitative research approach was used for the study which was appropriate for
the purpose and objectives of the study. The Authors DID NOT, however, specify the sampling technique and method used to select participants.
The statistical tests used for analyzing the study findings were appropriate for achieving the
objectives of the study.
The findings are relevant and important to Preventive health practices and contribute to
Knowledge.
The discussion involves mainly explaining and comparing the current findings with
existing literature.
The authors DID NOT outline the limitations of the study
The overall appearance of the manuscript (Typing, Organisation of content) was very
good, and the entire work is well organized. hence, it is appropriate for publishing.
There appeared to be enough citations to support the work, and the Authors followed the
guidelines specified by the journal. 

Round 2

Reviewer 1 Report

Comments and Suggestions for Authors

Thank you for your revisions and responses. It appears that the authors may have overlooked my initial suggestion to replace the term flora with microbiota throughout the manuscript. While flora was once commonly used, it is now considered outdated and inaccurate. After correcting this minor detail, I recommend the manuscript for publication.

Author Response

Reviewer comment - Thank you for your revisions and responses. It appears that the authors may have overlooked my initial suggestion to replace the term flora with microbiota throughout the manuscript. While flora was once commonly used, it is now considered outdated and inaccurate. After correcting this minor detail, I recommend the manuscript for publication.

Response - This was indeed an oversight, we have now replaced it throughout the manuscript.